# Rare Amyloid Precursor Protein Point Mutations Recapitulate Worldwide Migration and Admixture in Healthy Individuals: Implications for the Study of Neurodegeneration

**DOI:** 10.3390/ijms232415871

**Published:** 2022-12-14

**Authors:** Paolo Abondio, Francesco Bruno, Amalia Cecilia Bruni, Donata Luiselli

**Affiliations:** 1Laboratory of Ancient DNA, Department of Cultural Heritage, University of Bologna, Via degli Ariani 1, 48121 Ravenna, Italy; 2Laboratory of Molecular Anthropology and Center for Genome Biology, Department of Biological, Geological and Environmental Sciences, University of Bologna, Via Selmi 3, 40126 Bologna, Italy; 3Regional Neurogenetic Center (CRN), Department of Primary Care, ASP Catanzaro, 88046 Lamezia Terme, Italy; 4Association for Neurogenetic Research (ARN), 88046 Lamezia Terme, Italy

**Keywords:** amyloid precursor protein, Alzheimer’s disease, neurodegenerative diseases, neurodegeneration, dementia, population genomics, haplotypes, admixture, neuropathology, point mutation

## Abstract

Genetic discoveries related to Alzheimer’s disease and other dementias have been performed using either large cohorts of affected subjects or multiple individuals from the same pedigree, therefore disregarding mutations in the context of healthy groups. Moreover, a large portion of studies so far have been performed on individuals of European ancestry, with a remarkable lack of epidemiological and genomic data from underrepresented populations. In the present study, 70 single-point mutations on the *APP* gene in a publicly available genetic dataset that included 2504 healthy individuals from 26 populations were scanned, and their distribution was analyzed. Furthermore, after gametic phase reconstruction, a pairwise comparison of the segments surrounding the mutations was performed to reveal patterns of haplotype sharing that could point to specific cross-population and cross-ancestry admixture events. Eight mutations were detected in the worldwide dataset, with several of them being specific for a single individual, population, or macroarea. Patterns of segment sharing reflected recent historical events of migration and admixture possibly linked to colonization campaigns. These observations reveal the population dynamics of the considered APP mutations in worldwide human groups and support the development of ancestry-informed screening practices for the improvement of precision and personalized approaches to neurodegeneration and dementia.

## 1. Introduction

The amyloid precursor protein (APP) is a type I integral membrane glycoprotein that is formed by a large N-terminal glycosylated domain on the extracellular side and a smaller intracellular C-terminal domain. It is encoded by the *APP* gene located on the long arm of chromosome 21 [1]. There are several isoforms of APP, but the predominant ones in the human brain, sorted by aminoacidic chain length, are APP695, APP751, and APP770 [2,3]. The physiological role of APP is not yet fully understood and remains open to debate in the scientific community, but several studies suggest that the protein, as expressed in the brain, seems to be important in regulating neuron development and survival, synaptic function and plasticity, cell adhesion, and neuroprotection [3].

As shown in Figure 1, under physiological conditions, APP is subject to two processing pathways, namely, one nonamyloidogenic (α-secretase pathway) and one amyloidogenic (β-secretase pathway), the second of which leads to the production of damaging beta-amyloid (Aβ) peptides [4]. In the nonamyloidogenic pathway, the proteolytic cut of α-secretase on the extracellular side of APP generates the soluble fragment sAPPα into the extracellular space. The remaining intracellular CTFα/CTF83 is further cleaved by the γ-secretase complex, leading to the formation of p3 and AICD fragments, which are further processed by caspases to produce Jcasp and C31 [5]. This pathway is nonamyloidogenic as α-secretase cuts within the sequence of the otherwise damaging Aβ peptide [6].

On the other hand, in the amyloidogenic pathway, the proteolytic cleavage of APP by β-secretase occurs at the level of the extreme N-terminal of the Aβ sequence, resulting in the formation and subsequent release of the sAPPβ fragment into the extracellular domain and the formation of CTFβ/CTF99, which remains in the membrane. CTFβ is then cleaved by γ-secretase (a complex of four proteins, including presenilin, nicastrin, Aph-1, and Pen-2), which cuts in different positions at the C-terminal end of the Aβ sequence. The result of this processing is the formation of Aβ fragments between 39 and 43 amino acids in length (of which the most abundant are βA40 and βA42) that are secreted and the formation of the previously mentioned AICD fragments Jcasp and C31 [3,7,8].

To date, 52 pathogenic mutations in the *APP* gene that can lead to Aβ deposition in the brain parenchyma and in cerebral blood vessels and at least 26 missense mutations located within or immediately flanking the Aβ sequence have been linked to autosomal dominant Alzheimer’s disease (ADAD) [9,10]. Interestingly, the rare coding variant APP A673T has been detected in Norwegian, Swedish, Icelandic, and Finnish populations and reported as being associated with reduced risk for AD [9]. The same mutation is absent in healthy elderly individuals of Asian descent and is much rarer in the U.S. population [11,12,13]. As far as we know, however, no more general population genetic studies have been performed on the *APP* gene in healthy individuals. Indeed, it must be noted that the majority of the crucial genetic discoveries related to AD and other dementias have been performed using either large cohorts of affected subjects or multiple individuals from the same pedigree, therefore disregarding their existence in the context of healthy groups. However, it is well known that isoforms of the same protein (e.g., APOE2, APOE3, and APOE4 for apolipoprotein E) have different functions based on the tissue in which they are expressed (pleiotropic gene) and that the context determines varying expression of pathological conditions that are often linked to a protective function of these mutations in specific environments [14]. Moreover, a large portion of the studies so far have been performed on individuals of general European ancestry (and often excluding descendants of people from the Iberian Peninsula and Finland), with a remarkable lack of epidemiological and genomic data from the underrepresented South American, African, South Asian, and East Asian populations. This hinders the overall understanding of the impact of genetic mutations and their phenotypic manifestations in the context of neurodegeneration and dementia [15,16,17,18].

Given these premises, the present study was aimed at detecting the possible presence of 70 single-point mutations along the sequence of the *APP* gene in a publicly available dataset (identified as the “1KGP dataset”) that included the genetic sequences of 2504 healthy individuals from 26 worldwide populations in five macroareas. All mutations detected in the dataset were analyzed for their distribution across populations and macroareas. Moreover, a pairwise comparison of the segments surrounding the mutations was performed to reveal patterns of haplotype sharing that could point to specific cross-population and cross-ancestry admixture events, therefore shedding light on the population dynamics of the considered *APP* mutations in worldwide human groups. 

## 2. Results

### 2.1. Distribution of APP Point Mutations

When the selected 70 single-nucleotide variants for the *APP* gene (see Appendix A) were searched along the sequence of chromosome 21 in the entire 1KGP dataset that included 2504 individuals, a total of eight variants of interest were detected in 51 different individuals (30 females and 21 males; see Table 1 and Appendix A). As the requirement for this study was that an individual had to be at least heterozygous for the mutation, the other 62 variants along the APP sequence could not be further explored through this dataset, and the remaining 2453 individuals were removed. 

All mutation carriers (excluding the female TSI subject NA20259, who presented two copies of the KM670/671NL “Swedish” double mutation) were heterozygous, and two nonsynonymous mutations (which change the identity of the aminoacidic residue in the protein sequence) were never carried together by the same individual. However, synonymous mutations G708G and I716I were both found together in eight individuals representative of the African macroarea. Indeed, several mutations appeared to be limited to a single individual (A713T in a male Mexican carrier and V604M in a female Japanese subject), a single population (E599K in three Finnish individuals), or a single macroarea (the aforementioned “Swedish” double mutation in seven people of European descent), while others seemed to cross the boundaries of physical geography and population ancestry (c.*372A>G in three representatives from Europe, two from South Asia, and one admixed American; G708G in eight African subjects, four Europeans, and two people of Asian ancestry; and S614G in 16 representatives of the African macroarea as well as in three admixed Americans). Given the overall rarity of these mutations and the fact that this study, which focused on healthy individuals at the time of data collection, found a cumulative frequency for the mutations of interest at about 2%, there is the possibility that the supposedly deleterious alleles did not appear by chance in multiple individuals of the same population. Indeed, investigating the sharing of identical segments around the mutation for the carriers of that specific allele could reveal a common origin for each variant of interest instead of random genetic variation.

### 2.2. Segment Sharing across Individuals

For each of the eight detected mutations, segment sharing was investigated by comparing pairs of chromosome sequences among carriers and identifying a distribution of segment lengths that was then standardized as described in Section 4.5. Statistically significant segments of unusually large size and carrying the mutation of interest were detected for six of the APP aminoacidic variants (see Appendix A), with the exclusion of mutations A713T and V604M, for which comparative analysis with other carriers could not be performed.

When comparing chromosomes carrying the genetic mutation causative of the E599K aminoacidic change, the three subjects belonging to the Finnish population all exhibited pairwise significant segment sharing (Appendix A). In particular, the two male individuals shared a 227 kb segment spanning almost 6000 SNPs, while they both shared the same 107 kb region (including 2543 point variants) with the representative Finnish female. All other comparisons (between a chromosome copy carrying the mutation and one not carrying it or between two noncarrier chromosome copies) revealed much shorter segments (<44 kb) shared among pairs of individuals, and those that reached statistical significance were found either between the two male Finnish representatives or the chromosome copies of the female carrying or not carrying the mutation. According to the *gnomAD* genome aggregation database browser (version 2.1.1; https://gnomad.broadinstitute.org, last accessed 24 October 2022), the genetic substitution causative of E599K could be detected overwhelmingly in the European population (397 out of 422 times over a total of 282,872 observations, of which 210 times were in Finnish people) and never in East Asian individuals. Moreover, the frequency of this allele in the Finnish group (8.3 × 10^−3^) was almost six times that of the whole population considered by *gnomAD* (1.5 × 10^−3^). However, few studies have reported the specific mutation as causative of disease. In one study, it was found in one control of European descent (aged at least 60 at time of death and heterozygous for APOE4) among 179 healthy subjects when compared with 141 cases of late-onset AD [19]. Moreover, it was detected in subjects with Parkinson’s disease (PD) with and without associated dementia [20] as well as in two of 424 French individuals with early-onset AD [21]. Although initially suspected as possibly damaging [19], the mutation has since been classified as “nonpathogenic” [22].

Comparing carriers for the mutation c.*372A>G detected in the 3′ untranslated region of the *APP* gene, it could be seen that significant segments were shared indiscriminately among pairs of individuals of European and South Asian ancestry, especially when comparing either two carrier chromosomes (length ranging from 26 up to 647 kb) or two noncarrier chromosomes (even though the latter comparisons provided much shorter significant segments ranging from 14.7 up to 49 kb). The admixed Colombian individual representative of American ancestry (CLM) shared a limited, but significant, carrier segment of 15.4 kb with the Sri Lanka Tamil (STU) subject and, when comparing noncarrier and carrier chromosomes, an even shorter one (11 kb) with both the aforementioned STU individual and a European carrier of Iberian ancestry (Appendix A). The mutation has been already studied in the context of cerebral amyloid angiopathy (CAA), where it was detected in two cases but also in controls [23]. Interestingly, the individual who developed symptoms at a later age (73 years) was heterozygous for APOE4 and had no recorded family history of cerebral microbleeds or hemorrhage, while the younger subject (54 years) was homozygous for APOE4 and the mother had an intracerebral hemorrhage that could have been consistent with CAA [23], showing that other genetic factors may contribute to the phenotypic manifestations of this mutation.

Analysis of individuals with the synonymous protein sequence change G708G revealed that, compared to chromosomes carrying the genetic mutation, significantly long segments were present either between subjects of African ancestry (including the admixed Caribbean individual ACB) or people of European and South Asian descent, but no significant sharing was detected between African and non-African individuals or with the Vietnamese carrier KHV (see Appendix A). When chromosomes carrying the mutation were contrasted with chromosomes not carrying it, the noncarrier chromosome of the KHV subject did indeed share a significant DNA segment with the mutated chromosome of an Iberian individual, while the noncarrier chromosome of the ACB subject and that of the Bengali individual representative of South Asia (BEB) showed significant sharing with mutation-carrying chromosomes from the individuals of European descent (Appendix A). According to *gnomAD*, the genetic mutation responsible for this synonymous protein change had a frequency of 8.1 × 10^−3^ in the African population, four times more than non-Finnish Europeans (almost 2.0 × 10^−3^) or East Asians (1.7 × 10^−3^) and 20 times more than people of Finnish or South Asian ancestry (both 5.2 × 10^−4^). The mutation was identified in an early study [24] that included individuals from Southern Sweden, where it was identified in two of 12 AD patients, one of 60 patients without AD diagnosis, and one of 30 healthy controls. However, its possible pathogenicity was quickly dismissed by a subsequent analysis of 50 other individuals with AD from Central Sweden [25], where the mutation was not detected. Recently, a study of 404 Chinese pedigrees with familial AD revealed the mutation in a proband in association with a heterozygous condition for the APOE4 allele [26]. Accordingly, this mutation has been classified as a risk factor for AD rather than a direct cause [22].

The pattern of sharing for the mutation inducing the synonymous change I716I almost exactly mirrored that of G708G as both were found specifically in the same individuals of African ancestry. In pairwise comparisons, the admixed individual ACB shared shorter genetic segments carrying the mutation than the other subjects (see Appendix A). Indeed, seven of the individuals carried both mutations on the same chromosome copy, so the length of significant shared segments during pairwise comparison was exactly the same when considering the genetic surroundings of either mutation. However, the male individual belonging to the Luhya population from Kenya (LWK of identifier NA19383) carried the mutations G708G and I716I on different copies of chromosome 21. He shared the smallest significant segments when comparing sequences both carrying the I716I causative mutation, while the segments shared for the mutation G708G were not significant at all (Appendix A). Interestingly, no literature exists concerning this synonymous mutation, its population distribution, or its putative association with pathological conditions.

The genetic variation inducing the aminoacidic change S614G was carried predominantly by individuals of African ancestry. However, several significantly long segments carrying the mutation can be identified by comparing these sequences with those of the Peruvian (PEL) and, to a lesser degree, the Puerto Rican subject, both representatives of the American macroarea (see Appendix A). Similarly, when searching for shared segments between carrier and noncarrier chromosomes, the largest significantly long fragments were found between nonadmixed people of African ancestry and the admixed representatives of Southwestern USA (ASW), the Caribbean (ACB), and the American macroarea (PEL, PUR, and CLM). These segments, however, were all much shorter than the ones found when comparing two chromosomes carrying the mutation (see Appendix A). This mutation has been detected primarily in people of African ancestry as the *gnomAD* database reports its frequency at 1.5% in that specific cohort, which includes 376 out of a total of 404 mutated alleles from the whole dataset. The variant has been studied in one paper concerning familial dementias in Caribbean people of Iberian ancestry [27], where it was found in a single case as well as in seven unrelated controls.

The “Swedish” double-mutation KM670/671NL, found only in individuals of European ancestry, revealed that the longest significant segment (>181 kb) between carrier chromosomes was shared by a representative of Central European ancestry (CEU) and a British individual (GBR). Interestingly, the chromosome copies from the homozygous individual from Tuscany (TSI) shared a much smaller segment between each other (16.8 kb), and while one shared very large segments (126 and 96 kb) with noncarrier copies from a GBR and a CEU individual, respectively, the other only shared fragments equal to or smaller than 16.8 kb with any other noncarrier chromosome (see Appendix A). This mutation has been known since 1992 [28] and has been associated with cerebral atrophy as well as an increase in pathogenic βA40 and βA42 fragments in carriers [22]. However, an interesting effect has been recently noticed in vitro, where human cultured neurons expressing the mutation increased their number of synapses by 20 times in a process probably driven by the supposedly pathogenic fragments [29].

Figure 2 summarizes the putative origins and dynamics of the mutations described here.

## 3. Discussion

The present study was carried out to assess whether it was possible to detect genetic biomarkers of neurodegeneration (namely, single-nucleotide variants putatively associated to AD that would involve the sequence of the APP) in a dataset of healthy individuals representing worldwide genetic diversity. Indeed, some of these mutations were found, and the pattern of sharing for the surrounding genetic region was analyzed through pairwise sequence comparison among carrier individuals after gametic phase reconstruction and haplotype estimation were performed. Notably, each mutation detected in this study was uniquely shared by a group of individuals and, in general, two alleles of interest were never carried by the same subject (Appendix A). This suggests that these mutations, even in healthy individuals, do not tend to cluster or accumulate in the genetic sequence of *APP*, implying that the patterns of sharing for the different mutations can reveal separate instances of population dynamics. Taking advantage of notions from population genomics, it can be noted that several of the observed variants of interest presented distributions that can be justified either by the random appearance of the same mutation in different populations and geographic areas or by the movement of a genetic segment already carrying the mutation from a population to another through admixture or sometimes by the appearance of a mutation after a noncarrier genetic segment has passed from a population to another. Moreover, it appears as though these patterns replicate instances of historical migration that took place over the last millennium.

For example, it is known that, although rare, the mutation A713T found here in an admixed Mexican subject has already been identified with different manifestations (early-onset, late-onset, familial, and sporadic AD) in individuals or families from France [30,31], Spain [32], the United Kingdom [33], and Argentina [34]. Similarly, a major cluster was found when analyzing numerous unrelated families of Italian origin and ancestry (specifically from the Calabria region of Southern Italy [35]), where multiple generations had been affected by familial AD, often accompanied by cerebral amyloid angiopathy (CAA) and stroke as vascular manifestations of the disease [36,37,38]. Indeed, it has also been estimated that, given the size of the shared segments around it, this specific mutation may present a local Southern Italian origin from a common ancestor who lived there more than 1000 years ago [39]. People of Iberian descent have been historically associated with the colonization of Central and South America after the “European discovery of the New World”, during the Age of Discovery in the early 16th century [40]. It has not to be forgotten that sections of today’s Italian and French territories were also part of the Spanish Empire, so a non-negligible genetic input from non-Spanish European settlers of Mediterranean origin may have occurred at the time [41,42]. Similarly, it is estimated that during the so-called Italian diaspora (starting with the unification of Italy in 1861, with a third wave still occurring today), about 30 million people left the country, more than half of them before World War I (1914) [43,44,45]. The impact of this mass migration, which involved many Central and South American countries, can be observed through the largest communities of Italian descent existing today, including about 47% of people of Argentina (corresponding to roughly 19.7 million individuals), 35% of people of Uruguay (1.2 millions), and 15% of current Brazilians (32 millions), not to forget 17 million Americans of Italian ancestry who also live in the United States [40,44]. Therefore, alongside the plausible hypothesis of a random origin for the mutation A713T in the Mexican individual, there is the possibility that its admixed European ancestry may have provided the deleterious allele (Figure 1), although further studies are needed in order to prove it.

Mutation V604M, found here in a Japanese individual, has recently been identified in a total of three subjects from Thailand showing symptoms of early-onset AD [46,47,48]. Indeed, there is quite high genetic homogeneity among ethnic groups of Southeast Asian ancestry [49], so a recent common ancestor for the mutation may exist even though the two areas are geographically distant (Figure 2). It is also possible that contacts leading to local admixture have taken place since the 13th century, first in the form of piracy and, three centuries later, through highly regulated commercial trade routes in Asian waters under the Ryukyu kingdom of Okinawa, which helped establish Japanese shipping colonies on foreign land facing the South China Sea and adjoining gulfs [50,51,52].

The synonymous variant G708G suggests that a more complex dynamics is involved. Indeed, segments detected among pairs of carrier chromosomes indicated that there was no sharing pattern across macroareas, suggesting that the same mutation may have appeared separately in Africa, Europe, and East Asia at different times (Figure 1). This is not unexpected as a neutral mutation that does not change the sequence, function, or expression of a protein product may emerge multiple times in distinct populations, be preserved, and pass along subsequent generations without being lost. However, the sharing of a very large significant segment surrounding the variant of interest between the noncarrier chromosome of the KHV subject and the mutated chromosome of the Iberian individual also aligns historically with the arrival of the first Portuguese missionaries in Vietnam and other Southeast Asian countries at the beginning of the 16th century [53,54]. Therefore, in this case, a genetic segment may have been exchanged between the European settlers and the local communities around that time (Figure 1). A similar event may justify the sharing of a significantly long segment between the individual of Bengali ancestry and several European subjects if the British rule over the Indian subcontinent is also factored in. In fact, the British East India company, created in 1600 to manage East–West trade across the Indian Ocean, had exerted commercial, military, and administrative power over Southeast Asia until the company’s dissolution and the assumption of direct control by the British crown in 1858 [55]. Moreover, the admixed Caribbean subject (ACB) revealed a particularly intriguing story as its noncarrier chromosome showed significant sharing with mutation-carrying chromosomes from individuals of European descent, while the carrier chromosome shared a significant segment with other carrier chromosomes from African populations (Figure 1). Considering the aforementioned European colonial ventures of the 17th century (and, in particular, the French and British dominion over the “West Indies” after seizing them from Spain [56,57]) as well as the establishment of Jamaica as the center of the Anglo-American trade of African people across the Atlantic Ocean to use as slaves [58,59], it seems possible that at least one healthy Caribbean individual would show the effect of mixed ancestry in the different nature of her chromosome copies.

In this study, we also reported the presence of the synonymous variant I716I in only individuals of African descent. Moreover, in seven out of eight cases, they were found on the exact same segment also carrying mutation G708G; only one male individual (NA19383) showed the two mutations on two different chromosome copies. To our knowledge, no specific study exists on this mutation that can help in the interpretation of the result. The *gnomAD* browser indicates that the genetic change G>T causing the mutation is much more prevalent in people of African ancestry (186 out of 195 alleles detected over a sample size of 282,738 alleles), while the G>A genetic change is prevalent in non-Finnish European subjects (20 of 22 alleles over the previously mentioned sample size), which may explain why only African individuals with the mutation were found here. The limited number of studies on the genetics of dementia and neurodegeneration in African populations impairs the possibility to analyze this result further, but at the same time provides a great opportunity for its future investigation [60]. We would like to suggest, however, that the existence of two synonymous mutations at close distance to one another may present an effect on the APP protein, even though its aminoacidic sequence is not changed. Several studies have proposed that changes in the nucleic acid triplets composing a codon (the sequence of three nitrous bases that underpins the incorporation of aminoacidic residues in a protein) may modulate specific properties by changing the speed of protein synthesis [61,62]. Indeed, codon usage bias (i.e., the difference in frequency occurrence of synonymous base triplets in coding DNA) implies in general that some codons may be preferentially used during protein synthesis as they allow for optimization of translation rates with high accuracy [61,63,64] and that this is a mechanism to balance mutation bias with natural selection as seen in fast-growing organisms with relatively small genomes [62,65,66]. The same phenomenon has been recently verified for human pluripotent embryonic stem cells, where codon bias was related to the different guanine-cytosine (GC) content of differentially expressed genes during stem cell differentiation [67]. In the specific case of G708G and I716I mutations, it would be interesting to experimentally test if codon bias has an influence on either translation speed or accuracy and, given the vectorial nature of in vivo protein synthesis, if this in turn impacts phenomena such as protein folding, membrane insertion, and degradation.

## 4. Materials and Methods

### 4.1. Population Data Recovery

The distribution and variability of *APP* point mutations was analyzed in the publicly available Phase 3 dataset (hereby identified as “1KGP dataset”) produced by the 1000 Genome Project Consortium [68], which includes 2504 unrelated, self-reportedly healthy adult individuals (age > 18) representatives of 26 populations and classified in five macroareas (Africa, America, Europe, South Asia, and East Asia). The dataset comprises populations collected in diaspora (where population ancestry and geographical location of the samples do not match) and groups of admixed ancestry, thus offering the opportunity to investigate the cross-population and cross-ancestry genetic context in which the mutations exist.

### 4.2. APP Gene Point Mutations

Mutations for the *APP* gene studied in the context of neurodegeneration were recovered from ALZforum (https://www.alzforum.org/mutations/app; last accessed 10 September 2022), an open-access resource for research news and information about Alzheimer’s and related diseases that catalogs scientific data about genetic variants and their influence on the manifestation and modulation of pathological phenotypes. Specifically, it manages a repository of genetic variants in genes linked to AD with the goal of providing a list of variants that have been reported in the literature, ranging from causative to benign. For each variant, the salient clinical and neuropathologic features as well as the functional effects are reported. At last access, it included the three genes (APP, PSEN1, and PSEN2) associated with ADAD, plus two genes (MAPT and TREM2) with genetic associations to AD and related disorders. A total of 70 single-nucleotide variants involved in both synonymous and nonsynonymous aminoacidic changes of the canonical 770-residue-long APP protein sequence were recovered from the database (Appendix A); the mutations were checked for codon change sequence and NCBI dbSNP unique variant identifier (in the form rsX, where “X” is a unique numerical sequence; https://www.ncbi.nlm.nih.gov/snp/, accessed on 10 September 2022).

### 4.3. Dataset Quality Control

A standard quality control (QC) procedure was performed on the complete 1KGP dataset, which includes almost 80 million single-nucleotide variants (as identified on the hg19 reference human sequence) covering all autosomal chromosomes (therefore excluding the X and Y chromosomes as well as mitochondrial DNA). This was performed before extracting variants limited to chromosome 21, where the gene of interest is located, so as not to introduce any bias. QC was performed with the PLINK version 1.9 software (https://www.cog-genomics.org/plink/, accessed on 10 September 2022) [69] and included the removal of variants or individuals with data missingness greater than 1% (--geno 0.01; --mind 0.01). A check for the respect of Hardy–Weinberg equilibrium for each variant was also performed applying a Bonferroni correction for multiple testing to the standard threshold of 0.01, which was then divided by the number of variants (--hwe 1.8 × 10^−10^). Furthermore, ambiguous SNVs (carrying an A/T or C/G combination of alleles, for which the chromosome copy and strand cannot be univocally defined) were removed, and all remaining variants on chromosome 21 were extracted (--chr 21) for haplotype reconstruction.

### 4.4. Data Phasing and Haplotype Estimation

In order to assess the presence of the APP mutation in the context of the surrounding variants, detect finer-scale relationship patterns, and compare chromosomal sequences for further statistical analysis (see Section 4.5), a data-phasing procedure and haplotype estimation was performed in order to define which allele of every variant was located on which copy of the chromosome for each of the 2504 individuals of the 1KGP dataset. To do so, information about the ancestral or derived nature of each variant was deduced using a reconstructed reference human genome sequence as a guide to distinguish between ancestral and derived alleles. In particular, the ancestral/derived state of each allele in such a reference sequence was first assigned by aligning it with the Ensembl Compara 6 primates EPO genome sequences [70], and only alleles present in all the compared genomes were considered as ancestral. Haplotype estimation was finally performed using the SHAPEIT software version 1.9 (https://mathgen.stats.ox.ac.uk/genetics_software/shapeit/shapeit, accessed on 10 September 2022) [71,72] with default parameters settings and the HapMap phase 3 recombination map for chromosome 21. Reconstruction was performed on a total of 870,655 SNPs.

### 4.5. Segment Detection and Statistical Validation

Detection of shared chromosomal segments harboring the identified mutations in the 1KGP chromosome 21 dataset was performed separately for each single mutation and limited to the individuals who carry it. Pairs of chromosomes (both with the mutation or one with it and one without it) were compared point-by-point along their entire length through an ad hoc Python script (version 2.7.13; https://www.python.org, accessed on 10 September 2022), and a “segment” was considered as each continuous sequence of at least two alleles that were identical in the compared chromosome sequences. This generated a distribution of segment lengths, where “length” was defined as the absolute value of the difference in physical positions between the extremities of each segment. To statistically define if the shared segment carrying the mutation was significantly longer than the others (and, as such, could be considered uncommonly shared between individuals), the distribution of lengths obtained for each pairwise chromosome comparison was standardized by subtracting the average value of the distribution from each length and then dividing the result by the standard deviation of the same distribution. The new obtained distribution had the characteristics of a standardized distribution centered around the zero value, where the normalized value of each length indicates how many standard deviations each segment falls from the average of the distribution. Absolute normalized values above two standard deviations suggests that the associated segment falls within the extreme values of the distribution, thus indicating that the sharing of segments between two individuals with respect to all the other shared segments along the chromosome could be considered significant.

## 5. Conclusions

The present study, which involved the detection and distribution analysis of rare *APP* point mutations in a publicly available dataset that included individuals from 26 global populations representing human genetic variability, reveals that variants usually analyzed in the context of their pathological manifestations may indeed be recovered from large datasets of healthy subjects. Furthermore, significant sharing of chromosomal segments (haplotypes) surrounding the variants of interest reveals cross-population contact and peculiar cross-ancestry dynamics that can be at least partially traced back to the worldwide phenomena of migration, colonization, and admixture in recent history. Indeed, individual and population ancestry, driven by past and recent microevolutionary dynamics, are relevant for the presence of specific causative mutations in several human groups, so they cannot be discarded as factors determining the evaluation and contextualization of genetic variants associated with neurodegenerative diseases. Moreover, even if the individual is carrying a causative mutation, its penetrance and expression are modulated both by its genetic background and the environmental context, so the same mutation in different populations may not have the same phenotypic manifestation. Given the rate of both historical and contemporary worldwide migrations, this knowledge may be useful for the development of ancestry-informed screening practices and improvement of precision and personalized approaches to neurodegeneration and dementia.

However, the objective of this work and its usefulness extend beyond neurodegenerative disease screening and prevention, indicating the necessity of acquiring extensive knowledge around putatively causative mutations for which, very frequently, even their exact origin is unknown. Since the inception of paleoanthropology and the joint study of modern, ancient, and archaic humans, it has been revealed that several genetic changes associated with adaptive phenotypes and diseases in the current populations have been inherited from our ancestral cousins [73,74,75]. Moreover, it has become evident that the ancestral history of a population has a strong impact on the genetic makeup and, consequently, the phenotypes of current individuals in that population (see two recent examples in [76,77]). We also have to take into consideration the incredible extension of human lifespan, which has helped in revealing the neurodegenerative impact of mutations that could not have been known if the average age at death of a contemporary human was that of 1000 years ago and the fact that the human context in which mutations are expressed changes their effect. Therefore, it is advantageous to build a wealth of knowledge around the evolutionary and population history of human mutations in order to appropriately deal with such a complex phenomenon as neurodegeneration. This study points, for example, towards the potential usefulness of an ancestry-informed screening for specific point mutations that are different for different populations as it may be advantageous, especially for individuals from populations of historically admixed ancestry as well as for people whose recent ancestors have moved to a different country. 

We also acknowledge that having more individuals for each detected mutation would enormously support our observations as a larger set of tests, association analyses, and ancestry estimates could be performed to obtain very specific results. Sadly, this was impossible to do with the few data we could collect at the time of this study as the analysis of 51 individuals from all over the world would not generally yield results that are statistically strong enough to stand by themselves. Moreover, the 1000 Genome Project dataset is a collection of genetic sequences specifically intended for population genomic purposes, not for clinical or diagnostic use. Its aim is to represent most of the genetic variability of humans worldwide, irrespective of the age or phenotypic condition of the individuals. For the dataset to be fully publicly available, the consortium that created it could not collect any kind of family history, biometric data, or biological markers. The only requirements for inclusion in the project were being at least 18 years old and self-reportedly healthy. Of course, this implies that we could never know if any of the carriers have actually developed the disease over time or if other conditions are linked to these mutations, but that is not the main purpose of the paper. However, leveraging recent maritime and migration history has helped us in contextualizing our considerations and making sense of what the data are indeed suggesting. Moreover, this study presents itself as a proof of concept that different populations may carry distinct causative mutations, that these mutations may come from a different population entirely, and that care must therefore be taken in assuming generalized expectations during genetic screenings for neurodegenerative diseases, among others.

## Figures and Tables

**Figure 1 ijms-23-15871-f001:**
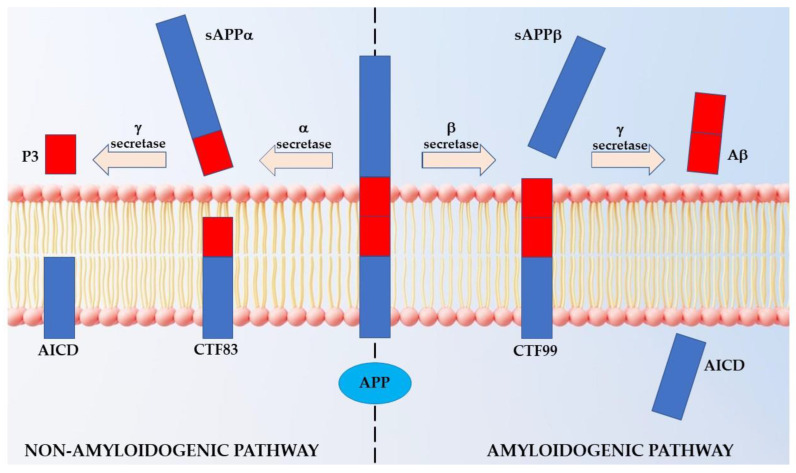
Amyloidogenic and nonamyloidogenic processing of the APP protein under physiological conditions. The region giving origin to the Aβ fragment is highlighted in red.

**Figure 2 ijms-23-15871-f002:**
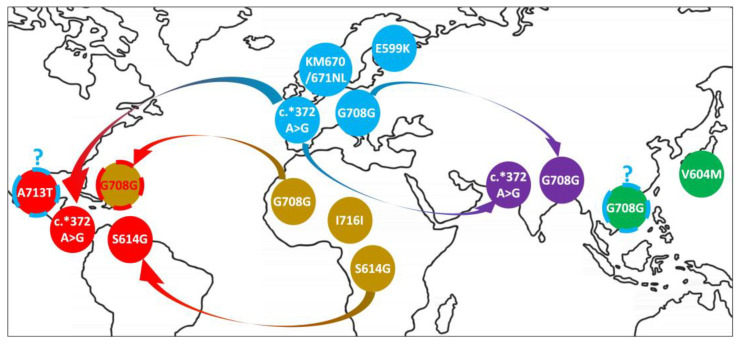
Possible origin and cross-population sharing of mutation-carrying segments. The Caribbean population of admixed origin is indicated by red text and red dashed borders. The putative European contributions to individuals carrying mutations A713T in America and G708G in Vietnam are indicated by a blue question mark and blue dashed borders.

**Table 1 ijms-23-15871-t001:** APP point mutations detected in the worldwide dataset.

Protein Mutation	Location	Recorded Codon Change	dbSNP_ID	Position (hg19)	Macroareas ^2^
c.*372 A>G	3′ UTR	Noncoding region	rs187940037	27253609	AMR, EUR, SAS
I716I ^1^	Exon 17	ATC to ATA or ATT	rs145564988	27264097	AFR
A713T	Exon 17	GCG to ACG	rs63750066	27264108	AMR
G708G	Exon 17	GGC to GGT	rs148888161	27264121	AFR, EAS, EUR, SAS
KM670/671NL	Exon 16	AAG.ATG to AAT.CTG	rs281865161	27269938	EUR
S614G	Exon 14	AGC to GGC	rs112263157	27284122	AFR, AMR
V604M	Exon 14	GTG to ATG	rs199887707	27284152	EAS
E599K	Exon 14	GAA to AAA	rs140304729	27284167	EUR

^1^ Mutation I716I was not present in the original 70 variants taken from the ALZforum database. ^2^ AFR: African; AMR: American; EAS: East Asian; EUR: European; SAS: South Asian.

## Data Availability

All data used in this study are in the public domain, and their sources are indicated in the main text; no new data were produced for this study.

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
