# Peer review of "Rare Amyloid Precursor Protein Point Mutations Recapitulate Worldwide Migration and Admixture in Healthy Individuals: Implications for the Study of Neurodegeneration"

_ijms, 2022, doi:10.3390/ijms232415871_

Round 1

Reviewer 1 Report

Unlike in previous similar investigations where the genetic database only limited to a few populations, this work performed a thorough genetic analysis abased on a large database with 2504 healthy individuals from 26 populations, aiming to identify some important mutation points relevant to neurodegenerative diseases of Alzheimer’s disease and other dementias. In general, this work is well planned and the results are convincing. I did not find any major problem. They have made some interesting discoveries, shedding light on the population dynamics of the considered APP mutations in worldwide human groups. I think this work could be published after the following improvements.

(1)     The significance of this work in preventing and screening of neurodegenerative diseases need to be further elaborated.

(2)     The supporting information, though mentioned in the main text, is lacking.

(3)     The English is good. Yet, there are some minor writing issues. For example, should 2.504 be 2,504? Line 358, -10 should be superscript. Doi of reference #1 is missing.

Author Response

C1. The significance of this work in preventing and screening of neurodegenerative diseases need to be further elaborated.

R1. We would like to thank the Reviewer for this comment. Indeed, the objective of this work and its usefulness does not directly link to neurodegenerative disease screening and prevention, but to the necessity to acquire extensive
knowledge around putatively causative mutations for which, very frequently, even their exact origin is unknown. Since the inception of paleoanthropology and the study of modern, ancient and archaic humans together, it has been revealed that several genetic changes associated to diseases in the current populations have been inherited from our ancestral cousins; moreover, it has become evident that the ancestral history of a population has a strong impact on the genetic makeup and, consequently, the phenotypes of current individuals in that population. We also have to take into consideration the incredible extension of human lifespan, which has helped in revealing the neurodegenerative impact of mutations that could not have been known, if the average age at death of a contemporary human was that of 1,000 years ago, and that the human context in which mutations are expressed changes their effect. So, it is advantageous to build a wealth of knowledge around the evolutionary and population history of human mutations in order to appropriately deal with such a complex phenomenon as neurodegeneration. This study points, for example, towards the potential usefulness of an ancestry-informed screening for specific point mutations, that are different for different populations, as it may be advantageous especially for individuals from populations of historically admixed ancestry, as well as for people whose recent ancestors have moved to a different country. All of this has been added to the “Discussion” section to further emphasize the significance of this work.

C2. The supporting information, though mentioned in the main text, is lacking.
R2. We thank the reviewer for highlighting this issue. We acknowledge that having many more individuals for each detected mutation would enormously improve the support for our observations, as a larger battery of tests, association
analyses and ancestry estimates could be performed to obtain very specific results. Sadly, this was impossible to do with the few data we could collect at the time of this study, as the analysis of 51 individuals from all over the world would
not yield statistically strong enough results to stand by themselves. However, leveraging recent maritime and migration history has helped us in contextualizing our considerations and making sense of what the data is indeed suggesting. Moreover, this study presents itself as a proof of concept that different populations may carry distinct causative mutations, that these mutations may come from a different population entirely and, therefore, that care must be taken in assuming generalized expectations during genetic screenings.

C3. The English is good. Yet, there are some minor writing issues. For example, should 2.504 be 2,504? Line 358, -10
should be superscript. Doi of reference #1 is missing.
R3. We thank the Reviewer for the observation. The whole text has been scanned thoroughly by the authors for writing issues. All thousands separators have been converted to commas, all decimal separators have been converted to points,
all powers are now superscript. All references have been double-checked for their completeness. Reference 1 does indeed not have a doi.

Reviewer 2 Report

Abondio et al analysed 70 APP point mutations from 51 healthy individuals from 26 populations using a publicly available database and analyzed their distribution. Eight mutations were detected in the worldwide dataset and pair wise comparisons of sections surrounding the mutations revealed 89 patterns of haplotype sharing that may signify specific cross-population and cross-ancestry admixture events worldwide. This is an important study as unlike previous studies, it focused on non-European countries, the details of which are less known in the context of mutations in APP and the links to neurodegeneration. Whilst this is an important preliminary study and the result are interesting,  I have some questions /points that need addressing before the work can be published.

Points for feedback.

It would be useful to have the methods section before the results section as it is difficult to analyse the results without knowing what data was used and how it was analysed.

Line 97 – ‘a total of eight variants of interest’ why were they of interest? More detail about these 8 point mutations needs to be summarised in the results section for the reader e.g. how are they linked to disease?

Lines 102-110 – This section is too text heavy and could be displayed better. The countries these mutations are found in could be added to the table? Or create a separate table? At present it is difficult for the reader to follow.

Line 190 - British/Scottish individual (GBR) this is confusing, just say British

Line 97-98 – this is a small study with only 51 participants. Why were so few individuals analysed? This needs to be explained here. Line  321 says ‘2.504’ does this refer to the number of individuals available in the database? This is not clear.  

Also, what is known about the family history of these individuals? We know that they are ‘healthy’ but do they have any other related conditions that may influence the results? How old are the participants? Be more specific than saying ‘>18’ as stated in line 322.

Discussion and conclusions- This is a small sample size, be careful to not overinterpret the results. You need to state in these sections why only a small number of participants were involved in this study and state that further studies using more individuals (more than 51) needs to be done to obtain statistically significant results. Also in the conclusions you need to explain the impact of this work, how will this data be used now? Why is it useful?

Supplementary data- Lots of important information is only available in supplementary data. Tables 3 and 5 for example contain important data- can this be summarised and displayed as an additional figure in the main text?

Author Response

C1. It would be useful to have the methods section before the results section as it is difficult to analyse the results
without knowing what data was used and how it was analysed.
R1. We thank the Reviewer for this observation. We strictly followed the manuscript structure as provided by the
journal, so we cannot simply put the “Methods” section before the “Results” section. However, we have added several
sentences throughout the text, hoping that the reader may engage more easily with the material.
C2. Line 97 – ‘a total of eight variants of interest’ why were they of interest? More detail about these 8 point mutations
needs to be summarised in the results section for the reader e.g. how are they linked to disease?
R2. We would like to thank the Reviewer for asking more details. Indeed, all 70 single nucleotide variants recovered
from the ALZforum database were “variants of interest” from the start, as they all have already been studied in the
context of Alzheimer’s disease, either familial or sporadic. The eight ones that were found also in the 1000 Genome
Project dataset were of particular interest, because they could help us in highlighting specific patterns of population
dynamics. As suggested by the Reviewer, we have added details about the pathology of each mutation in the “Results”
section, with the exclusion of mutations A713T and V604M, which have been already contextualized in the “Discussion”
section.
C3. Lines 102-110 – This section is too text heavy and could be displayed better. The countries these mutations are
found in could be added to the table? Or create a separate table? At present it is difficult for the reader to follow.
R3. We thank the Reviewer for highlighting this issue. While we agree that this part of the manuscript is not as agile as
others, we specifically decided to write down individual numerosity for each mutation, because we found out that such
information reported into a table in the main text would be cumbersome. As suggested, we did modify Table 1 to include
a new column at the end, reporting the macroareas each mutation appears into. We would also like to point out that,
as made explicit at the beginning of that paragraph, extended information on the identity of each individual and their
population of origin, in relation to each carried mutation, can be found in Supplementary Table S2.
C4. Line 190 - British/Scottish individual (GBR) this is confusing, just say British
R4. We would like to thank the Reviewer for this note. Indeed, this was an oversight on our part: according to the
definitions provided at http://ftp.1000genomes.ebi.ac.uk/vol1/ftp/README_populations.md, the population code GBR
stands for “British (individuals) in England and Scotland”, that was inadvertently shortened into “British/Scottish
individual” instead of “English/Scottish individual”. Now, the mistake has been amended by simply using the word
“British”, as suggested by the Reviewer.
C5. Line 97-98 – this is a small study with only 51 participants. Why were so few individuals analysed? This needs to be
explained here. Line 321 says ‘2.504’ does this refer to the number of individuals available in the database? This is not
clear.
R5. We thank the Reviewer for pointing this out. We understand that this is a small-scale study with a limited number
of individuals. Indeed, the first step in this study was to detect which of the 70 selected APP mutations could also be
found in any of the 2,504 individuals belonging to the 1000 Genome Project. Eight of those mutations have been
identified in 51 individuals, which means that the other 62 mutations could not be further explored through this dataset,
and that the remaining 2,453 individuals could not be informative in contextualizing the population dynamics of those
eight mutations. This has now been specified in the main text.
C6. Also, what is known about the family history of these individuals? We know that they are ‘healthy’ but do they have
any other related conditions that may influence the results? How old are the participants? Be more specific than saying
‘>18’ as stated in line 322.
R6. We would like to thank the Reviewer for this comment. The 1000 Genome Project dataset is a collection of genetic
sequences specifically intended for population genomic purposes, not for clinical or diagnostic use. Its aim is to
represent most of the genetic variability of humans worldwide, irrespective of age or phenotypic condition of the
individuals. For the dataset to be fully publicly available, the Consortium that created it could not collect any kind of
family history, biometric data or biological markers. The only requirements for inclusion in the Project were being at
least 18 years old and self-reportedly healthy. Of course, this implies that we could never know if any of the carriers has
actually developed the disease or if other conditions are linked to these mutations, but that is not the focus of the paper.
C7. Discussion and conclusions- This is a small sample size, be careful to not overinterpret the results. You need to state
in these sections why only a small number of participants were involved in this study and state that further studies using
more individuals (more than 51) needs to be done to obtain statistically significant results. Also in the conclusions you
need to explain the impact of this work, how will this data be used now? Why is it useful?
R7.We would like to thank the Reviewer for the suggestion. We have added sentences explaining the limitations of this
study and the expectations of usefulness for this work in the “Conclusions” section.
C8. Supplementary data- Lots of important information is only available in supplementary data. Tables 3 and 5 for
example contain important data- can this be summarized and displayed as an additional figure in the main text?
R8. We would like to thank the Reviewer for this comment. Indeed, the Supplementary data are quite rich and
interesting, but very difficult to summarize. However, we already tried to provide an overview of the dynamics that can
be traced through the Supplementary Tables in the map presented in what is now Figure 2.

Reviewer 3 Report

Dear Editor,

Dear Authors,

 Thank you for inviting me to review the manuscript by Paolo Abondio et al., that covers an interesting topic but, in my opinion, requires a major revision:

11. Please do not use abbreviations as keywords. Keywords should facilitate the association with MeSH terms.

22. Spelling and grammar errors are still included in the manuscript.

33. Sections “Introduction” and “Discussion” should be written based on the very recently papers published on that subject. Please revise the cited papers and include current papers from the last 4 years.

44. Please include in the “Introduction” section a figure showing two processing pathways of the amyloid precursor protein.

Author Response

C1. Please do not use abbreviations as keywords. Keywords should facilitate the association with MeSH terms.

R1. We thank the Reviewer for highlighting this point. Abbreviations in the keywords have been removed and, after scanning the whole manuscript with the “MEsH on Demand” tool (provided by the NIH and available on https://meshb.nlm.nih.gov/MeSHonDemand), we have come up with an improved list of keywords that incorporate MEsH terms appropriately.

C2. Spelling and grammar errors are still included in the manuscript.

R2. We would like to thank the Reviewer for the observation. The whole text has been scanned thoroughly by the authors for writing issues.

C3. Sections “Introduction” and “Discussion” should be written based on the very recently papers published on that subject. Please revise the cited papers and include current papers from the last 4 years.

R3. We thank the Reviewer for pointing this out. Indeed, while most of the literature presented for this paper is quite recent, we do acknowledge that some of the papers in the “Introduction” section are older, either because of the seminal importance of the presented material or because the latest study on a specific topic of interest is indeed relatively old. Similarly, the books and scholarly monographs that contextualize our reasoning in the “Discussion” section often deal with very specialized topics in several fields (colonial history, human migration, politics and economics, maritime and naval history among others), so that it is extremely difficult to find hyper-current references. However, we did take the Reviewer’s suggestion into consideration.

C4. Please include in the “Introduction” section a figure showing two processing pathways of the amyloid precursor protein.

R4. We thank the reviewer for this note. Indeed, the suggested figure has been included in the “Introduction” section of the main text as “Figure 1”.
